# Interleukin 6 is increased in preclinical HNSCC models of acquired cetuximab resistance, but is not required for maintenance of resistance

**Rachel A. O'Keefe** [ID]**, Neil E. Bhola, David S. Lee, Daniel E. Johnson, Jennifer R. Grandis***

Department of Otolaryngology–Head and Neck Surgery, University of California San Francisco, San Francisco, CA, United States of America

* jennifer.grandis@ucsf.edu

## Abstract

The epidermal growth factor receptor inhibitor cetuximab is the only oncogene-targeted agent that has been FDA approved for the treatment of head and neck squamous cell carcinoma (HNSCC). Currently, there are no biomarkers used in the clinic to predict which HNSCC tumors will respond to cetuximab, and even in tumors that regress with treatment, acquired resistance occurs in the majority of cases. Though a number of mechanisms of acquired resistance to cetuximab have been identified in preclinical studies, no therapies targeting these resistance pathways have yet been effectively translated into the clinic. To address this unmet need, we examined the role of the cytokine interleukin 6 (IL-6) in acquired cetuximab resistance in preclinical models of HNSCC. We found that IL-6 secretion was increased in PE/CA-PJ49 cells that had acquired resistance to cetuximab compared to the parental cells from which they were derived. However, addition of exogenous IL-6 to parental cells did not promote cetuximab resistance, and inhibition of the IL-6 pathway did not restore cetuximab sensitivity in the cetuximab-resistant cells. Further examination of the IL-6 pathway revealed that expression of *IL6R*, which encodes a component of the IL-6 receptor, was decreased in cetuximab-resistant cells compared to parental cells, and that treatment of the cetuximab-resistant cells with exogenous IL-6 did not induce phosphorylation of signal transducer and activator of transcription 3, suggesting that the IL-6 pathway was functionally impaired in the cetuximab-resistant cells. These findings demonstrate that, even if IL-6 is increased in the context of cetuximab resistance, it is not necessarily required for maintenance of the resistant phenotype, and that targeting the IL-6 pathway may not restore sensitivity to cetuximab in cetuximab-refractory HNSCC.

## Introduction

Head and neck squamous cell carcinoma (HNSCC) arises in the oral cavity, larynx, and pharynx and is the sixth most common cancer worldwide [1]. Each year, over half a million patients are diagnosed with HNSCC, and despite aggressive therapy, the five-year survival rate for patients with this cancer has hovered around 50% for decades [1–5]. Although the prescribed

**Funding:** This work was supported by National Institutes of Health grants R35 CA231998 (JRG), R01 DE024728 (DEJ), and F31 DE026951 (RAO). The funders had no role in study design, data collection and analysis, decision to publish, or preparation of the manuscript.

**Competing interests:** DEJ and JRG are coinventors of cyclic STAT3 decoy and have financial interests in STAT3 Therapeutics. STAT3 Therapeutics holds an interest in cyclic STAT3 decoy, which was not used in the studies in this manuscript. The remaining authors declare no conflicts. This does not alter our adherence to PLOS ONE policies on sharing data and materials.

treatment for HNSCC varies based on tumor stage, anatomical location, and disease etiology, chemotherapy, radiation, and/or surgery are commonly employed [4–11]. In addition, a number of targeted therapies have been or are currently being tested in clinical trials, and two anti-PD-1 immune checkpoint inhibitors were approved for treatment of HNSCC by the United States Food and Drug Administration (FDA) in 2016. To date, the only FDA-approved oncogene-targeted therapy is cetuximab (Ctx; Erbitux), a monoclonal antibody that is directed against the epidermal growth factor receptor (EGFR), which is overexpressed in up to 90% of HNSCC tumors [3,8,12–14]. Despite extensive evidence supporting EGFR as a therapeutic target in HNSCC, the response rate for single-agent cetuximab is below 20% in this disease [15], and therapy-resistant tumors arise in the majority of cetuximab-treated HNSCC patients [16]. Identification and targeting of pathways that mediate intrinsic and acquired cetuximab resistance could augment the effectiveness of treatment with this drug.

An ongoing challenge in the treatment of HNSCC is the lack of a predictive biomarker(s) for response to cetuximab. Because cetuximab targets EGFR, it was initially hypothesized that high tumoral expression of EGFR would identify tumors most likely to respond to cetuximab; however, this has not proven to be the case [17–19]. In colorectal cancer, another malignancy for which cetuximab has been FDA approved, tumors harboring activating mutations in *KRAS*, *NRAS*, or *BRAF* are often resistant to cetuximab, and mutations in these genes serve as a negative predictive biomarker for cetuximab response [17,20,21]. In HNSCC, activating mutations in *KRAS*, *NRAS*, and *BRAF* are rarely observed, and though activating mutations in *HRAS* have been implicated in cetuximab resistance in HNSCC [22,23], *HRAS* alterations are observed in only 5% of HNSCC tumors and its robustness as a negative predictive biomarker remains unproven [6]. Identification of a predictive biomarker(s) could improve outcomes by enabling identification of a subset of patients whose tumors are likely (or unlikely) to respond to cetuximab. To this end, a recent Phase II clinical trial sought to identify serum biomarkers that could predict resistance to a combination of cetuximab and the Src family kinase inhibitor dasatinib in HNSCC patients whose tumors had previously progressed on cetuximab-containing therapy [17]. In an analysis of four candidate serum biomarkers, only interleukin (IL)-6 levels were shown to be correlated with resistance to the combination of cetuximab and dasatinib [17], identifying high serum IL-6 as a potential predictive biomarker of resistance to cetuximab.

This study was not the first to identify a potential role for IL-6 in HNSCC. IL-6 is a pleiotropic cytokine that has been shown to be a negative prognostic factor in HNSCC [24]. In addition, previous investigations have demonstrated that IL-6 and downstream mediators of IL-6 signaling, particularly signal transducer and activator of transcription 3 (STAT3) and the phosphoinositide 3-kinase/Akt (PI3K/Akt) pathway, can play a role in resistance to EGFR-targeted therapies [13,25,26]. IL-6, therefore, is a potential predictive biomarker of cetuximab resistance in HNSCC and a plausible therapeutic target, particularly in the context of cetuximab-resistant disease. Acquired resistance to cetuximab remains a major obstacle in the effective treatment of HNSCC. Even in tumors that initially respond to cetuximab-containing therapy, activation of tumor cell-intrinsic and -extrinsic mechanisms can lead to cetuximab resistance. To date, these resistance mechanisms are poorly understood and no strategies to overcome cetuximab resistance have been translated to the clinic. However, drugs targeting IL-6 and the IL-6 receptor, as well as downstream components of the IL-6 pathway, although not FDA approved for the treatment of HNSCC, have been FDA approved for other indications [27–29].

Based on the results of the Phase II trial of cetuximab and dasatinib in HNSCC [17], as well as preclinical evidence supporting a role for IL-6 in cetuximab resistance, we investigated the role of the IL-6 pathway in preclinical HNSCC models of acquired cetuximab resistance. We hypothesized that IL-6 would promote cetuximab resistance in HNSCC cells and that inhibiting the IL-6 pathway in a cell line model of acquired cetuximab resistance would restore

sensitivity to cetuximab. Instead, we found that, despite increased IL-6 secretion in our cetuximab-resistant (Ctx$^R$) models, treatment of the parental (cetuximab-sensitive) cells with exogenous IL-6 did not promote cetuximab resistance, nor did inhibition of components of the IL-6 pathway restore cetuximab sensitivity in the Ctx$^R$ cells. Further, we found that expression of *IL6R*, which encodes the IL-6 receptor subunit IL-6Rα, was substantially reduced in the Ctx$^R$ cells compared to parental cells, and that Ctx$^R$ cells did not respond to IL-6 stimulation with an increase in phosphorylation of STAT3 at tyrosine 705 (Y705). Thus, though IL-6 secretion is correlated with cetuximab resistance in the PE/CA-PJ49 Ctx$^R$ models, IL-6 is not required for the maintenance of cetuximab resistance in these cells, and targeting the IL-6 pathway may not restore cetuximab sensitivity even in cetuximab-resistant tumors that exhibit increased expression of this cytokine.

## Results

### IL-6 secretion is increased in cell line models of acquired cetuximab resistance

We recently reported the generation of cell line models of acquired cetuximab resistance derived from the parental HNSCC cell lines PE/CA-PJ49, Cal33, and FaDu [30]. We reported that in these cell lines, an increase in expression of alternative receptor tyrosine kinases (RTKs), including AXL and MET, promoted resistance to EGFR inhibition. Expression of these alternative RTKs was driven by upregulation of the transcriptional co-activator bromodomain-containing protein-4 (BRD4), and targeting BRD4 was able to restore cetuximab sensitivity in these cells [30]. Whether additional alterations in these model cell lines contribute to cetuximab resistance remains unexplored. We found that treatment of the PE/CA-PJ49 parental and Ctx$^R$ cells with JQ1, an inhibitor of the bromodomain and extra-terminal (BET) family proteins (including BRD4), reduced *IL6* mRNA expression (**S1 Fig**). Thus, we utilized these models to assess the role of IL-6 in acquired resistance to cetuximab. We elected to focus on the PE/CA-PJ49 models because FaDu cells do not express the receptor subunit gp130, which is required for IL-6 signal transduction, and because, though *IL6* mRNA expression was slightly increased in the Cal33 Ctx$^R$ models compared to parental Cal33 cells, IL-6 levels in the Cal33 parental and Ctx$^R$ cell culture supernatants were very low (~25–50 pg/mL).

We first confirmed that these PE/CA-PJ49 Ctx$^R$ cells maintained resistance to cetuximab in 96-hour dose-response (**Fig 1A**) and 12-day clonogenic survival (**Fig 1B**) assays. Of note, PE/CA-PJ49 Ctx$^R$ cells were also resistant to the EGFR-targeted tyrosine kinase inhibitor (TKI) erlotinib and the dual EGFR/HER2-targeted TKIs afatinib and lapatinib (**S2 Fig**), but remained sensitive to cisplatin and CBL0137, a novel anti-cancer agent targeting the facilitates chromatin transcription (FACT) complex [31,32] (**S3 Fig**), suggesting cross-resistance to EGFR targeting but not general treatment resistance.

Previous studies focusing on mechanisms of resistance to cetuximab and other EGFR-targeted therapies have demonstrated that EGFR inhibitor-resistant cells secrete increased levels of IL-6 compared to sensitive cells [13,25,33]. Consistent with these reports [13,25,33], *IL6* mRNA expression was increased in the PE/CA-PJ49 cetuximab-resistant (Ctx$^R$) cells compared to the parental cells from which they were derived (**Fig 1C**). To measure secreted IL-6 in our models of acquired cetuximab resistance, PE/CA-PJ49 parental and Ctx$^R$ cells were plated in serum- and antibiotic-free media and an enzyme-linked immunosorbent assay (ELISA) was performed on cell culture supernatants collected after 72 hours. As shown in **Fig 1D**, IL-6 was increased in the cell culture supernatants of Ctx$^R$ cells compared to parental cells. These results suggested that IL-6 might play a role in acquired resistance to cetuximab in the PE/CA-PJ49 Ctx$^R$ models and provided the impetus for further investigation of the IL-6 pathway.

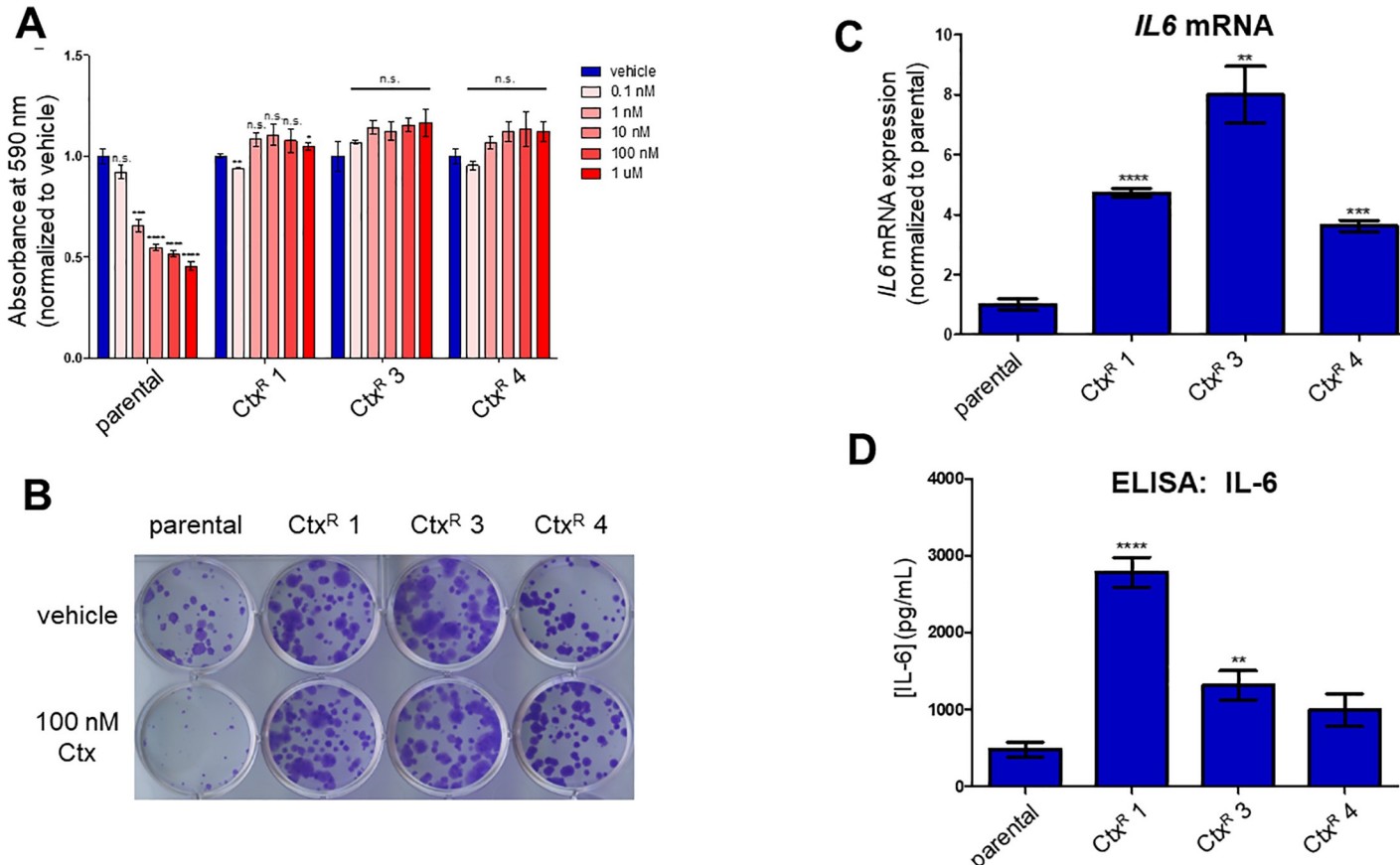

**Fig 1. Cell line models of acquired cetuximab resistance exhibit increased IL-6 secretion. A)** PE/CA-PJ49 parental and Ctx$^R$ cells were treated for 96 h with vehicle (PBS) or cetuximab (0.1 nM– 1 μM), then stained with crystal violet. Student's two-tailed t-test was used to determine whether differences in absorbance at 590 nm were statistically significant (compared to vehicle-treated cells). $n$ = 4. **B)** Cells were plated at low density and treated the next day with vehicle (PBS) or 100 nM cetuximab. Cells were stained with crystal violet after 12 days of cetuximab treatment. Media containing vehicle or cetuximab was changed every 4 days. **C)** RNA was extracted from PE/CA-PJ49 parental and Ctx$^R$ cells and qPCR was conducted using the *IL6* primers listed in **S1 Table** (normalized to *TBP*). $n$ = 3. **D)** PE/CA-PJ49 parental cells and Ctx$^R$ cells were plated in serum-free medium. Conditioned medium was collected after 72h and concentration of IL-6 was measured using ELISA. Student's two-tailed t-test was used to determine whether differences in *IL6* expression and secreted IL-6 were statistically significant (compared to parental cells). $n$ = 4. *p<0.05; **p<0.01; ***p<0.001; ****p<0.0001; n.s., not significant.

## Recombinant IL-6 does not confer cetuximab resistance in parental PE/CA-PJ49 cells

Because we observed an increase in IL-6 secretion in the cetuximab-resistant PE/CA-PJ49 cells compared to parental cells, and because IL-6 and its downstream effector STAT3 have been previously implicated in cetuximab resistance [13,25], we sought to determine whether addition of recombinant human IL-6 (rhIL6) would abrogate the growth inhibitory effects of cetuximab in PE/CA-PJ49 parental cells.

Before testing the impact of rhIL6 addition on cetuximab response in the PE/CA-PJ49 parental cells, we first determined whether rhIL6 was able to activate signaling downstream of the IL-6 receptor. The media on PE/CA-PJ49 parental cells was replaced with DMEM (no FBS) or DMEM supplemented with 10% FBS. After 4 hours, the cells were treated with 50 ng/mL rhIL6 for 15 minutes or 4 hours. Phosphorylation of STAT3 at tyrosine 705 (P-STAT3$^{Y705}$) was increased upon addition of rhIL6 in both the no FBS and 10% FBS conditions, while total STAT3 levels remained stable (**Fig 2A**), demonstrating that rhIL6 can indeed activate the JAK/STAT pathway in these cells even in the presence of serum. To assess the impact of rhIL6 on

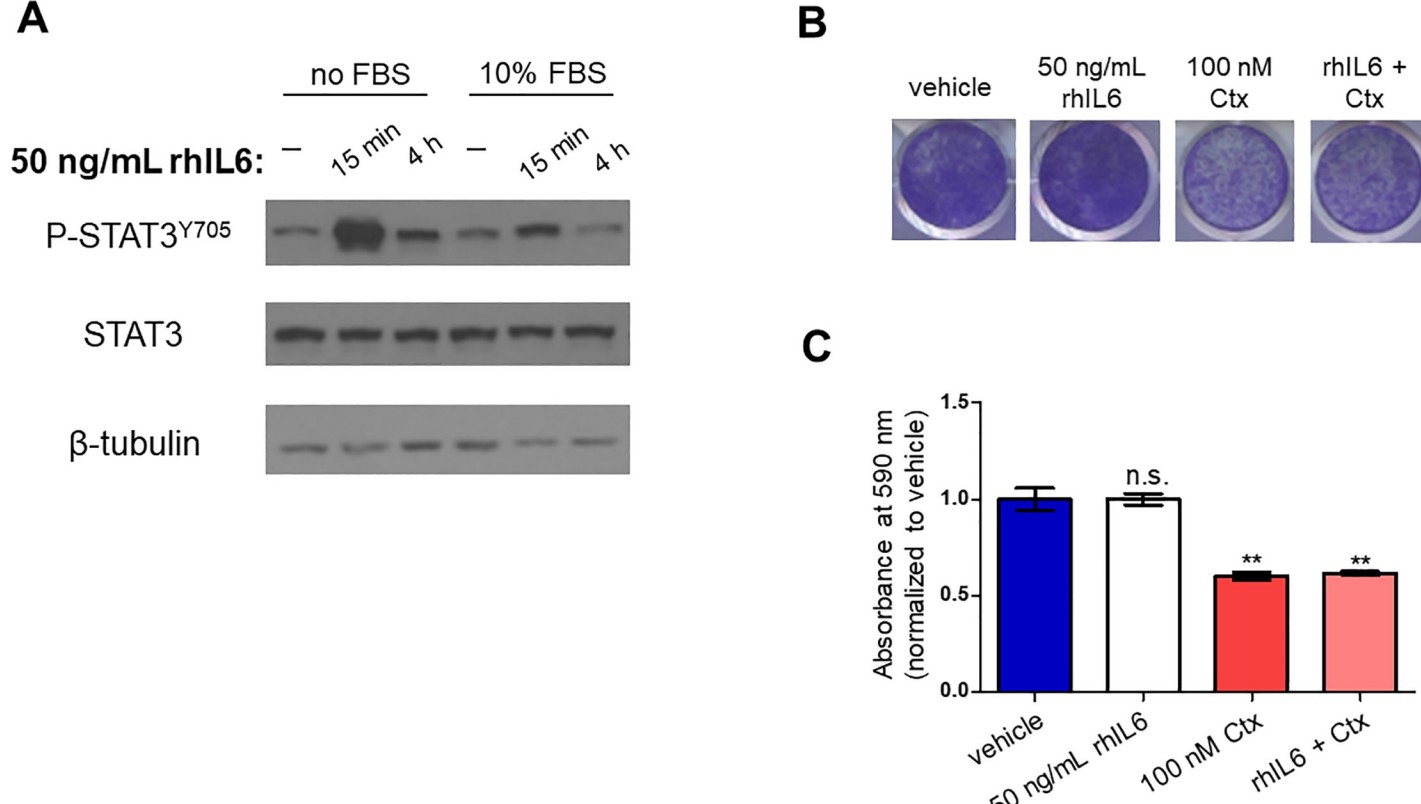

**Fig 2. Addition of recombinant IL-6 does not promote cetuximab resistance in PE/CA-PJ49 parental cells. A)** PE/CA-PJ49 parental cells were serum starved for 4 hours (no FBS) or remained in media containing 10% FBS (10% FBS), then treated with 50 ng/mL rhIL6 for 15 min or 4 hours. Cells were lysed in RIPA buffer and immunoblot was performed as described in Materials and Methods. β-tubulin image shown is from the STAT3 blot. **B)** PE/CA-PJ49 parental cells were treated for 96 h with vehicle (PBS), 50 ng/mL rhIL6, 100 nM Ctx, or the combination of Ctx and rhIL6, then stained with crystal violet. Images shown are representative of three biological replicates. **C)** Quantification of crystal violet staining in Fig 2B. Student's two-tailed t-test was used to determine whether differences in absorbance at 590 nm were statistically significant (compared to vehicle-treated cells). $n = 3$. ** $p < 0.01$; n.s., not significant.

cetuximab response, we treated the cells for 96 hours with 100 nM Ctx, 50 ng/mL rhIL6, or the combination of Ctx and rhIL6, then stained the cells with crystal violet (Fig 2B). The addition of rhIL6 did not prevent cetuximab-induced growth inhibition in the PE/CA-PJ49 parental cells (Fig 2B and 2C). These findings indicate that exogenous IL-6 alone cannot promote cetuximab resistance in this model cell line.

## Inhibition of the IL-6 pathway does not impact cetuximab response in PE/CA-PJ49 parental and Ctx$^R$ cells

Although addition of recombinant IL-6 did not promote cetuximab resistance in PE/CA-PJ49 parental cells, this did not rule out the possibility that IL-6 played a role in the maintenance of cetuximab resistance in the Ctx$^R$ cell lines. This was an appealing prospect because if IL-6 were required to maintain cetuximab resistance, then targeting the IL-6 pathway could be used to restore cetuximab sensitivity in these cell lines, and, potentially, in cetuximab-resistant HNSCC tumors. To determine whether inhibiting the IL-6 pathway could restore cetuximab sensitivity in the PE/CA-PJ49 Ctx$^R$ cell lines, we used both genetic and pharmacologic approaches to inhibit components of the IL-6 pathway alone and in combination with cetuximab.

To determine the impact of *IL6* knockdown on cetuximab response, we first confirmed that the siRNAs targeting *IL6* reduced *IL6* mRNA expression by transfecting PE/CA-PJ49 parental

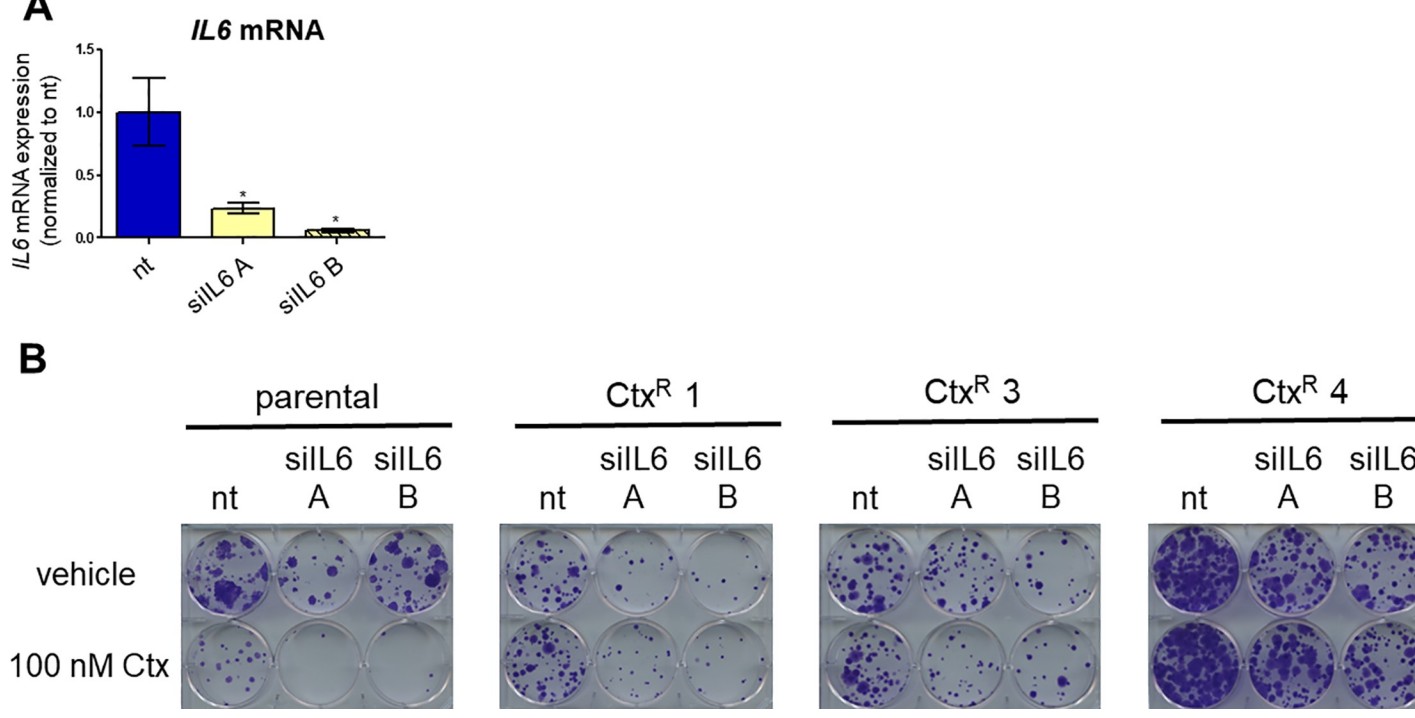

**Fig 3. Inhibition of IL-6 signaling does not impact cetuximab response in PE/CA-PJ49 parental and Ctx$^R$ cells. A)** PE/CA-PJ49 parental cells were transfected with 10 nM nontargeting (nt) siRNA or one of two siRNAs targeting *IL6* (siIL6 A and B). RNA was extracted 96 hours post-transfection and qPCR was conducted using the *IL6* primers listed in **S1 Table** (normalized to *TBP*). $n = 3$. *p<0.05. **B)** PE/CA-PJ49 parental and Ctx$^R$ cells were plated at a low density and transfected with 10 nM siRNA the next day. On the following day, and every four days thereafter, the cells were treated with vehicle (PBS) or 100 nM Ctx. The cells were stained with crystal violet 13 days post-transfection.

cells with 10 nM nontargeting (nt) siRNA or one of two distinct *IL6*-targeted siRNAs (siIL6 A and B). After 96 hours, quantitative reverse transcription PCR (qPCR) was performed to assess *IL6* mRNA levels. Cells transfected with either siIL6 A or siIL6 B exhibited a substantial reduction in *IL6* mRNA levels compared to nt-transfected cells (**Fig 3A**).

We next plated PE/CA-PJ49 parental and Ctx$^R$ cells at a low density to conduct clonogenic survival assays. Cells were transfected with 10 nM nt siRNA, siIL6 A, or siIL6 B and treated the next day (and every four days thereafter) with vehicle (PBS) or 100 nM Ctx. After 12 days of treatment (13 days post-transfection), the colonies were stained with crystal violet. As expected, PE/CA-PJ49 parental cells were sensitive to cetuximab (**Fig 3B**). Transfection with siIL6 itself reduced the number of colonies per well in the parental and Ctx$^R$ cells, and the combination of cetuximab and siIL6 had an additive effect in parental cells. However, siIL6-transfected Ctx$^R$ cells treated with vehicle and cetuximab were indistinguishable, demonstrating that the Ctx$^R$ cells remain resistant to cetuximab even when *IL6* expression is greatly reduced.

Next, we examined the impact of knocking down other components of the IL-6 pathway on cetuximab response. IL-6 signals through a receptor complex consisting of interleukin-6 receptor alpha (IL-6Rα, encoded by *IL6R*) and glycoprotein 130 (gp130, encoded by *IL6ST*). IL-6 signaling is initiated when IL-6 binds to IL-6Rα and the IL-6/IL-6Rα complex binds to gp130. Subsequent dimerization of gp130 leads to the formation of a heterohexameric signaling complex that recruits JAK proteins, leading to phosphorylation and nuclear localization of STAT3 [29,34–37]. Both IL-6Rα and gp130, in addition to IL-6, are required to initiate IL-6 signaling [29,35,36]; thus, if IL-6 signaling is required to maintain cetuximab resistance in the PE/

CA-PJ49 Ctx$^R$ cells, inhibition of either co-receptor would be expected to restore cetuximab sensitivity in these cells. Transfection of PE/CA-PJ49 parental cells with siIL6R (S4 Fig) or siIL6ST (S5 Fig) substantially reduced the mRNA levels of their respective targets in PE/CA-PJ49 parental cells. However, as observed when *IL6* was knocked down, neither siIL6R nor siIL6ST restored cetuximab sensitivity in the Ctx$^R$ cell lines (S4 and S5 Figs).

To corroborate the results we obtained using siRNAs with a more clinically relevant agent, we used tocilizumab (TCZ), an IL-6Rα-targeted monoclonal antibody that is FDA approved for the treatment of rheumatoid arthritis and chimeric antigen receptor (CAR) T cell-induced cytokine release syndrome. To select a concentration of TCZ for use in subsequent experiments, we serum starved PE/CA-PJ49 parental cells for 2 hours, then pre-treated the cells with vehicle (PBS) or increasing concentrations of TCZ for 2 hours before treating the cells for 15 minutes with 50 ng/mL rhIL6. We found that 100 nM TCZ was sufficient to block rhIL6-induced STAT3 phosphorylation (S6 Fig) and selected 1 μM TCZ as the concentration for subsequent experiments because a further decrease in STAT3 phosphorylation was observed in cells treated with this concentration.

To assess the impact of TCZ on cetuximab response in the PE/CA-PJ49 parental and Ctx$^R$ cells, we again plated the cells at a low density, then treated the cells with vehicle (PBS), 100 nM Ctx, 1 μM TCZ, or the combination of Ctx and TCZ, replacing media plus drug(s) every four days, for a total of 12 days of treatment (S6 Fig). In cells treated with cetuximab alone, a substantial decrease in crystal violet-stained material was observed in PE/CA-PJ49 parental cells, but not Ctx$^R$ cells. Treatment with TCZ, whether alone or in combination with cetuximab, did not have an impact on colony formation in parental or Ctx$^R$ cells (S6 Fig).

Despite a substantial increase in IL-6 levels (both mRNA expression and secreted cytokine) in the Ctx$^R$ cell lines compared to PE/CA-PJ49 parental cells, inhibition of components of the IL-6 pathway using both genetic (siRNA) and pharmacological (TCZ) methods did not impact cetuximab response in the Ctx$^R$ cells. Together, these results suggest that IL-6 signaling is not required for maintenance of cetuximab resistance in these models.

## Expression of components of the IL-6 pathway are altered in HNSCC cells that have acquired resistance to cetuximab

Though IL-6 levels are increased in PE/CA-PJ49 Ctx$^R$ cells compared to parental cells (Fig 1), treatment of parental cells with rhIL6 did not promote cetuximab resistance (Fig 2), and inhibition of the IL-6 pathway failed to reverse cetuximab resistance in the Ctx$^R$ cells (Fig 3; S4–S6 Figs). Seeking an explanation for this discrepancy, we analyzed expression of gp130 and IL-6Rα, both of which are required for IL-6 signal transduction, in parental and Ctx$^R$ PE/CA-PJ49 cells.

Expression of gp130 (encoded by the *IL6ST* gene) was evaluated by qPCR and immunoblot analysis and found to be increased at both the mRNA (Fig 4A) and protein (Fig 4C and 4D) levels in Ctx$^R$ cells compared to parental PE/CA-PJ49 cells. In contrast, mRNA expression of *IL6R* (the gene encoding IL-6Rα) was decreased in PE/CA-PJ49 Ctx$^R$ cells compared to parental cells (Fig 4B). This raises the question of whether IL-6 signaling is functionally intact in the PE/CA-PJ49 Ctx$^R$ cells.

## IL-6 signaling is impaired in PE/CA-PJ49 Ctx$^R$ cells

Although both IL-6 and gp130 levels were increased in Ctx$^R$ PE/CA-PJ49 cells, IL-6Rα levels were decreased in these cells compared to parental cells, revealing a disconnect among the components of the IL-6 signaling pathway. This discrepancy led us to examine the net impact of these alterations on downstream components of the IL-6 signaling pathway in the PE/CA-PJ49 Ctx$^R$ cells.

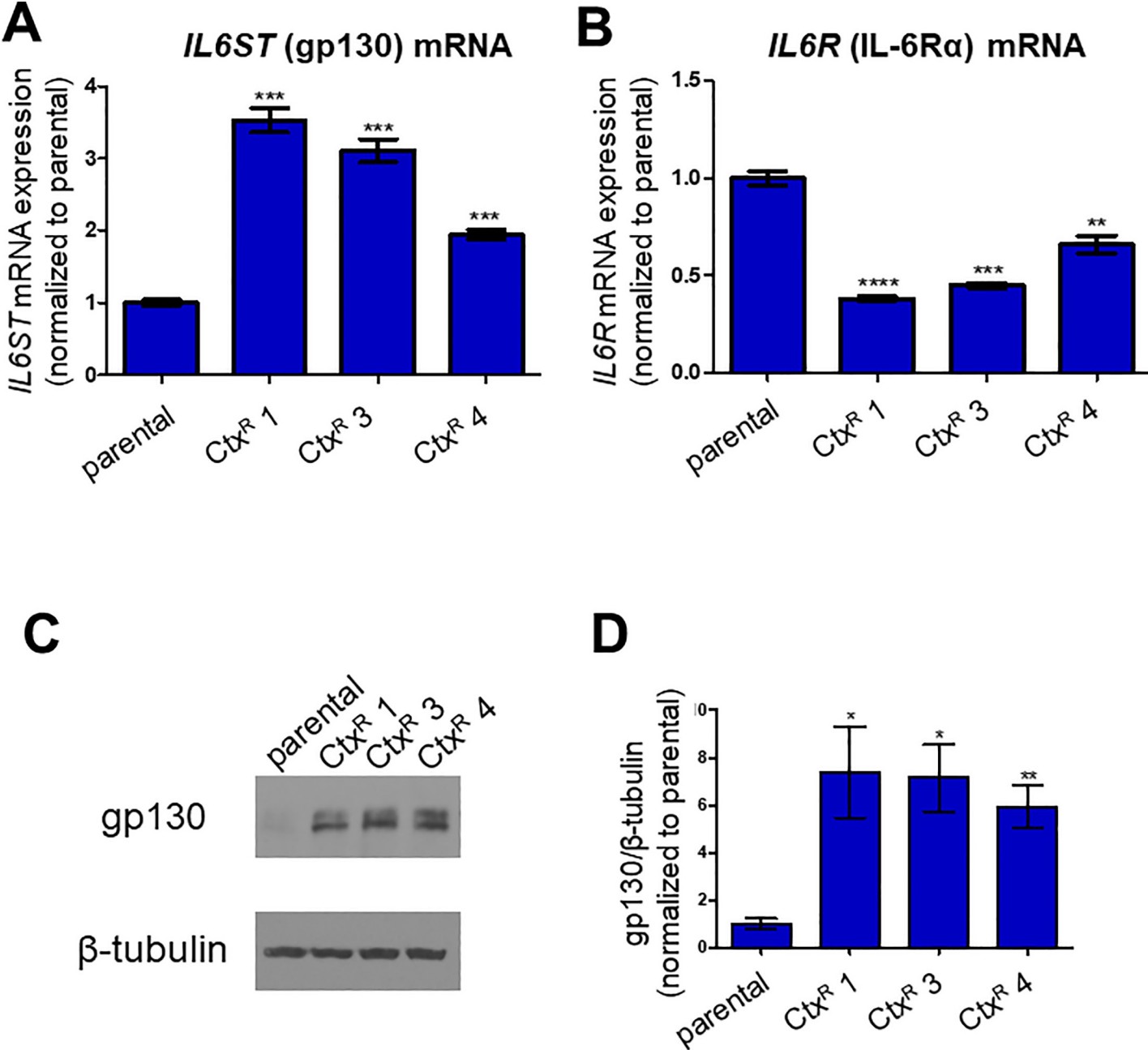

**Fig 4. Expression of components of the IL-6 pathway are altered in HNSCC cells that have acquired resistance to cetuximab. A,B)** RNA was extracted from PE/CA-PJ49 parental cells and cetuximab-resistant clones and qPCR was conducted using the *IL6ST* (**A**) or *IL6R* (**B**) primers listed in **S1 Table** (normalized to *TBP*). $n = 3$. **C)** Cells were lysed in RIPA buffer and immunoblot was performed. Images depicted are representative of three biological replicates. **D)** Densitometry was performed on the blots depicted in (**C**) using ImageJ as described in Materials and Methods. Densitometry values for gp130 were normalized to those for the loading control (β-tubulin). Student's t-test was used to determine whether differences in the gp130/β-tubulin ratios in Ctx$^R$ cells were statistically significant compared to parental cells. $n = 3$. $^*p<0.05$; $^{**}p<0.01$; $^{***}p<0.001$; $^{****}p<0.0001$.

We compared P-STAT3$^{Y705}$ levels in PE/CA-PJ49 parental and Ctx$^R$ cells and found that the ratio of P-STAT3$^{Y705}$ to total STAT3 was decreased in Ctx$^R$ cells compared to parental cells (**Fig 5A and 5B**), consistent with impaired IL-6 signaling in the Ctx$^R$ cells. However, because STAT3 phosphorylation is dynamically regulated by a number of kinases and phosphatases,

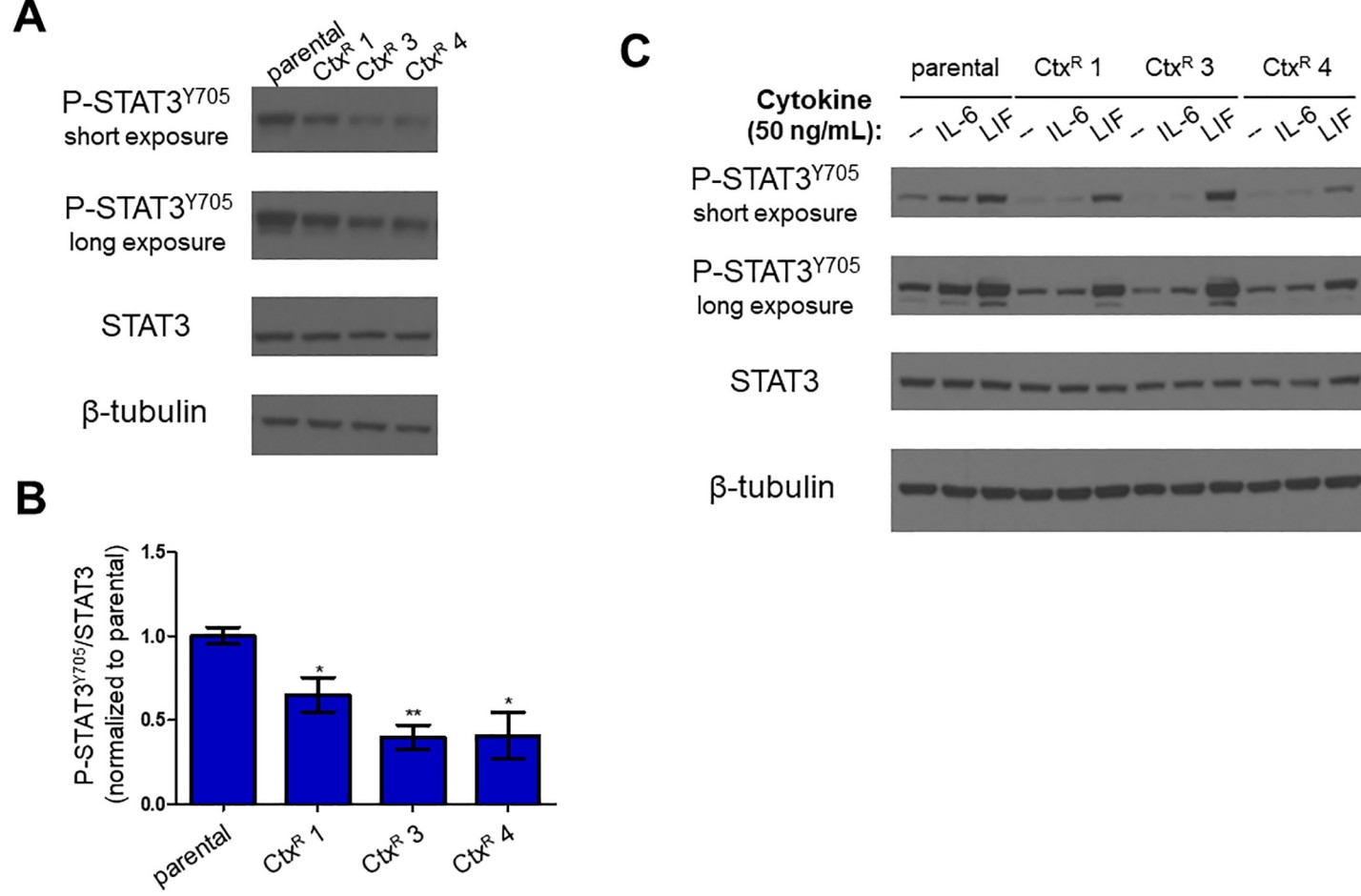

**Fig 5. IL-6 signaling is impaired in PE/CA-PJ49 Ctx$^R$ cells. A)** PE/CA-PJ49 parental and Ctx$^R$ cells were lysed in RIPA buffer and immunoblot was performed. Images shown are representative of three biological replicates. β-tubulin image shown is from the STAT3 blot. **B)** Densitometry was performed using ImageJ as described in Materials and Methods. Densitometry values for P-STAT3$^{Y705}$ were normalized to those for total STAT3. Student's t-test was used to determine whether differences in the P-STAT3$^{Y705}$:STAT3 ratios in Ctx$^R$ cells were statistically significant compared to parental cells. $n = 3$. $^*p<0.05$; $^{**}p<0.01$. **C)** PE/CA-PJ49 parental and Ctx$^R$ cells were serum starved for 4 hours, then treated for 15 minutes with 50 ng/mL rhIL6 or rhLIF. Cells were lysed in RIPA buffer and immunoblot was performed. β-tubulin image shown is from the P-STAT3$^{Y705}$ blot.

the decrease in P-STAT3$^{Y705}$ in the Ctx$^R$ cells alone does not conclusively demonstrate a defect in IL-6 signaling. Thus, we assessed IL-6 signaling more directly by serum starving parental and Ctx$^R$ PE/CA-PJ49 cells, then treating the cells for 15 minutes with 50 ng/mL rhIL6. Consistent with our findings in **Fig 2A**, addition of rhIL6 stimulated phosphorylation of STAT3 at Y705 in the parental cells. However, no increase in P-STAT3$^{Y705}$ was observed following 15 minutes of rhIL6 treatment in any of the Ctx$^R$ lines (**Fig 5C, S7 Fig**).

To determine whether this defect was IL-6-specific, we tested whether P-STAT3$^{Y705}$ levels in the Ctx$^R$ cells were increased following treatment with recombinant human leukemia inhibitory factor (rhLIF) or recombinant human oncostatin M (rhOSM). LIF and OSM are IL-6 family cytokines that utilize gp130 for signal transduction, but bind to non-IL-6Rα co-receptors to initiate signaling [35]. We serum starved PE/CA-PJ49 parental and Ctx$^R$ cells for 4 hours, then treated the cells for 15 minutes with 50 ng/mL rhIL6 or rhLIF. While rhLIF treatment induced STAT3 phosphorylation at Y705 in the parental and Ctx$^R$ cell lines, rhIL6 increased P-STAT3$^{Y705}$ levels in only the parental cells (**Fig 5C**). Similar results were observed

when cells were treated with rhOSM: Serum-starved parental and Ctx$^R$ cell lines all responded to rhOSM treatment with an increase in P-STAT3$^{Y705}$ (S7 Fig).

PE/CA-PJ49 parental and Ctx$^R$ cell lines all responded to treatment with the IL-6 family cytokines LIF and OSM with an increase in P-STAT3$^{Y705}$, demonstrating that gp130 and JAK/STAT3 signaling are functionally competent in these cells. In contrast, only the parental cells responded to treatment with rhIL6, suggesting that the decreased *IL6R* expression in the PE/CA-PJ49 Ctx$^R$ cells impacted their ability to mediate IL-6 signaling and providing further evidence that IL-6 is not required for the maintenance of cetuximab resistance in these cells.

## PE/CA-PJ49 parental and Ctx$^R$ cells exhibit distinct mutational profiles

Given our findings indicating that IL-6 is not required for maintenance of acquired cetuximab resistance in the PE/CA-PJ Ctx$^R$ cells, we elected to use an unbiased approach, next-generation DNA sequencing using a targeted gene panel (UCSF500 Cancer Gene Panel) [38], to identify potential mediators of cetuximab resistance in these models. In this panel, approximately 500 cancer-associated genes are sequenced, revealing single nucleotide variants (SNVs), insertions/deletions (indels), and copy number changes, as well as rearrangements that are commonly observed in cancer. This unbiased analysis could reveal novel genetic alterations correlated with acquired cetuximab resistance.

Genomic DNA isolated from the PE/CA-PJ49 parental and Ctx$^R$ cells was sent to the UCSF Clinical Cancer Genomics Laboratory for sequencing. The data generated from the analysis (S2–S5 Tables) were filtered as described in the Materials and Methods and used to compare the PE/CA-PJ49 parental and Ctx$^R$ cells. Many of the SNVs and indels were shared among the parental and Ctx$^R$ cells (Table 1), providing further evidence (along with short tandem repeat analysis) that the Ctx$^R$ cells are indeed PE/CA-PJ49 variants. However, there were several SNVs and indels that were observed in only one of the cell lines (Table 2). The parental cells harbored more unique SNVs and indels than any of the three Ctx$^R$ cell lines, but each cell line bore mutations that were not observed in the other three lines, suggesting heterogeneity among the clones (Table 2).

Notably, the analysis did not uncover any alterations in *EGFR* in the parental or the Ctx$^R$ cells, suggesting that alteration of the gene that encodes the protein targeted by cetuximab is

**Table 1. SNVs and indels identified in all PE/CA-PJ49 parental and Ctx$^R$ cell lines.**

| | |
|---|---|
| ARID1A p.A54S | NOTCH1 p.P1730L |
| ARID1A p.F1457S | NOTCH1 p.R365C |
| BRCA2 p.I247V | PAK1 p.E74D |
| CDKN2A p.M52fs | PEX11B p.G62V |
| CLPTM1L p.316_317del | PRDM1 p.R192C |
| COL1A1 p.P823A | PTPRD p.I1821V |
| CREBBP p.S1761* | RASA1 c.829_840GTAGAAGATAGA |
| FANCG c.176-2A>G | RASA1 p.A804fs |
| FAT1 c.11049_11050TT | RASA1 p.D280delinsDR |
| FLCN p.A90S | RASA2 p.Q286* |
| IPMK p.S261P | SYNE1 p.R4152C |
| JAK3 p.L1047V | TERT promoter |
| KMT2D p.M1478fs | TSC2 p.R1268C |
| † MIR4457 promoter | † ZFHX p.1823_1823del |

Exceptions (†) were observed in only the parental and Ctx$^R$ 3 cells. Abbreviations: del, deletion; fs, frameshift mutation.

**Table 2. SNVs and indels unique to individual cell lines.**

| Parental | CtxR 1 | CtxR 3 | CtxR 4 |
|---|---|---|---|
| • CHD1 p.K347R<br>• CHD5 p.101_102del<br>• EMSY p.E74K<br>• GNAQ p.V340F<br>• MTOR p.R2443Q<br>• MYH9 p.D1293N<br>• MYH9 p.E1270K<br>• NFKBIA p.E40K<br>• PDGFRA p.Y136fs<br>• PLCB4 p.A849P | • KAT6A p.1228_1228del<br>• NSD1 p.G1132fs | • CHD5 p.D774E<br>• HNF1A c.864delinsCC | • EIF1AX p.K56N<br>• PIK3R2 c.700_702CGT<br>• PRKDC exonic UNKNOWN<br>• RB1 p.W563*<br>• ZFHX4 p.2007_2007del |

Abbreviations: del, deletion; delins, deletion-insertion; fs, frameshift mutation.

not what mediates cetuximab resistance in these cells. In addition, although we observed increased gp130 levels in the CtxR cells, no alterations in the *IL6ST* gene were identified in the analysis (*IL6* and *IL6R* are not among the genes sequenced on this platform).

## Discussion

Based on evidence in preclinical models and the finding that serum IL-6 was a biomarker of resistance to cetuximab-containing therapy in a Phase II trial [17], we initially hypothesized that IL-6 mediated cetuximab resistance in HNSCC cells and that targeting the IL-6 pathway could overcome cetuximab resistance. Our initial characterization of *IL6* expression and secretion in the PE/CA-PJ49 CtxR cells was consistent with a role for IL-6 in cetuximab resistance, as IL-6 levels were increased in all three CtxR models compared to the parental PE/CA-PJ49 cells from which they were derived. However, inhibition of the IL-6 pathway did not restore cetuximab sensitivity in the CtxR models, and subsequent analyses revealed that, though parental PE/CA-PJ49 cells responded to IL-6 treatment with an increase in P-STAT3$^{Y705}$, the CtxR cells failed to do so, possibly due to the substantial decrease in expression of *IL6R* in the CtxR cells. These cumulative results suggest that, despite an increase in IL-6 secretion in the PE/CA-PJ49 CtxR models, IL-6 does not mediate cetuximab resistance in these models.

Our conclusion that IL-6 does not promote cetuximab resistance in the PE/CA-PJ49 CtxR cells, though initially surprising, is not without precedent. A previous study found that, despite increased secretion of IL-6 in cell line models of acquired resistance to cisplatin, IL-6 did not mediate cisplatin resistance in these cells [39]. Moreover, expression of *IL6R* was decreased in the cisplatin-resistant cells, and the authors speculated that this decrease in *IL6R* expression impaired IL-6 signaling [39]. Thus, in both cetuximab-resistant and cisplatin-resistant HNSCC cells, an increase in IL-6 secretion in conjunction with the emergence of drug resistant cells does not necessarily demonstrate that IL-6 is required for maintenance of drug resistance. It also raises the intriguing question of why *IL6R* expression was decreased in both of these drug resistance models. Was *IL6R* simply a bystander lost during acquisition of drug resistance, or were the decreases in *IL6R* expression due to selective pressure? Examination of these and other possibilities could be addressed in future studies.

The inability of the PE/CA-PJ49 CtxR cells to activate STAT3 in response to treatment with recombinant IL-6 suggests that the decrease in *IL6R* expression had a functional impact on response to IL-6. Because both IL-6Rα and gp130 are required to form functional IL-6 receptors, it is perhaps not surprising that the substantial reduction in IL-6Rα levels in PE/CA-PJ49 CtxR cells would impede the ability of IL-6 to induce STAT3 phosphorylation (**Fig 5C**; **S7 Fig**). In light of these findings, the inability of IL-6 pathway inhibition to reverse cetuximab

resistance in the Ctx$^R$ cells is not surprising; indeed, the results would have been difficult to interpret had IL-6 inhibition restored cetuximab sensitivity in cells in which IL-6 signaling is impaired. However, this finding may be model-specific, as IL-6 may mediate cetuximab resistance in cells in which the IL-6 signaling pathway is functional. Moreover, the inability of recombinant IL-6 to activate IL-6 signaling in the Ctx$^R$ cells does not rule out the possibility that IL-6 signals intracellularly, as IL-6 has been shown to activate signaling within endosomes [40]. This may explain an apparent discrepancy in **Fig 3** and **S4 and S6 Figs**, in which siRNA-mediated knockdown of *IL6* or *IL6R*, but not treatment with the IL-6Rα-targeted agent TCZ, reduces colony number in PE/CA-PJ49 parental and Ctx$^R$ cells. Though this does not appear to play a role in the maintenance of cetuximab resistance, since knockdown of *IL6*, *IL6R*, and *IL6ST* did not sensitize the cells to cetuximab, the respective contributions of intracellular and extracellular IL-6 in HNSCC may be a topic of further study.

Notably, our results do not rule out a potential role for IL-6 in cetuximab resistance in an *in vivo* setting. PE/CA-PJ49 cells do not reliably form xenograft tumors in even the severely immunocompromised NOD *scid* gamma strain of mice, so the experiments described above were conducted exclusively in cell culture or in samples derived from cell lines. These isolated cell culture models lack 3D architecture and the tumor microenvironment, including immune cells, which are especially relevant to the mechanism of action of cetuximab. Cetuximab is a monoclonal antibody and has been shown to mediate antibody-dependent cell-mediated cyto-toxicity (ADCC), a phenomenon in which immune cells (primarily natural killer cells) recognize and kill cells opsonized by antibodies [11]; thus, impairment of this process could impede the ability of cetuximab to promote immune-mediated tumor cell destruction. An abundance of evidence has established that IL-6 plays many roles in the tumor microenvironment, often as an immunosuppressive cytokine [41,42], and it is tempting to speculate that IL-6 secretion by cetuximab-resistant tumor cells could promote cetuximab resistance by downregulating cetuximab-induced ADCC. Future studies may explore whether IL-6 plays a role in resistance to cetuximab-induced ADCC, perhaps in a tumor/immune cell co-culture model.

Our findings also do not rule out the possibility that IL-6 was involved in the acquisition of cetuximab resistance in the PE/CA-PJ49 Ctx$^R$ models. In this study, we compared parental PE/CA-PJ49 cells that have never been exposed to cetuximab to PE/CA-PJ49 cells that have acquired resistance to cetuximab following months of treatment with increasing concentrations of the drug. Though we concluded that IL-6 is not required for the maintenance of cetuximab resistance in these cells, IL-6 could have played a role in the acquisition of cetuximab resistance. It has previously been shown that erlotinib-induced NF-κB-mediated *IL6* expression promotes survival of non-small cell lung cancer cells treated with this EGFR tyrosine kinase inhibitor, and it was suggested that this adaptive response to erlotinib treatment could ultimately promote acquired resistance to this drug [43]. Similarly, Fletcher and colleagues reported that erlotinib induced expression of *IL6* and other proinflammatory cytokines in HNSCC cells, and that combining erlotinib with the IL-6Rα-targeted antibody TCZ led to substantial reductions in tumor volumes compared to treatment with either drug alone in mouse xenograft models [44]. We observed an increase in *IL6* mRNA expression when parental PE/CA-PJ49 cells were treated with cetuximab for 96 hours (**S8 Fig**), raising the questions of whether IL-6 enabled the acquisition of cetuximab resistance in the PE/CA-PJ49 models and whether targeting the IL-6 pathway may delay or prevent the acquisition of cetuximab resistance. These questions may be a topic of future study.

Targeted sequencing of our models identified a number of unique mutations in each of the four cell lines tested. Notably, the PE/CA-PJ49 parental cells contained more unique SNVs and indels than any of the Ctx$^R$ cell lines, perhaps because each Ctx$^R$ cell line was derived from a single clone of cetuximab-treated parental cells. However, each of the Ctx$^R$ cells also

harbored unique mutations. Their absence in the parental cells suggests that these mutations arose *de novo* following the initiation of cetuximab treatment, but it remains possible that these mutations existed in the pool of PE/CA-PJ49 parental cells from which the Ctx$^R$ cells were derived, but were not present in a sufficient fraction of the parental cells to be detectable in our targeted sequencing analysis. Whether the mutations in the Ctx$^R$ cells are simply correlated with or indeed promote cetuximab resistance remains unknown. However, given previously published information on the functions of the proteins encoded by these genes, we can speculate on the potential roles of some of these mutations in cetuximab resistance.

Both of the genes that were altered in only the PE/CA-PJ49 Ctx$^R$ 1 cells, *KAT6A* and *NSD1* (**Table 2**), are involved in epigenetic regulation of gene expression. *KAT6A* encodes lysine acetyltransferase 6A (KAT6A), a histone acetyltransferase [45,46]. The specific alteration identified in this analysis (p.1228_1228del) has not been reported; thus, its impact on KAT6A protein expression and function are unknown. However, *KAT6A* mRNA levels are increased in glioblastoma samples compared to normal brain tissue, and KAT6A and has been shown to promote glioma cell proliferation via upregulation of *PIK3CA* expression and subsequent activation of PI3K/Akt signaling [45]. Thus, if the mutation identified in the Ctx$^R$ 1 cells is an activating mutation, this *KAT6A* alteration could promote cetuximab resistance by activating PI3K/Akt signaling, a previously identified mediator of cetuximab resistance [3,14,22,26]. *NSD1* encodes nuclear receptor binding SET domain protein 1 (NSD1), a histone methyltransferase [6,47,48]. The *NSD1* mutation identified in the Ctx$^R$ 1 cells, p.G1132fs, has not been previously reported. However, novel inactivating mutations in *NSD1* were identified in 29 of the 279 HNSCC tumors in the TCGA published in 2015 (all in HPV-negative tumors) [6], and the constellation of non-hotspot mutations suggest that this mutation, too, may be inactivating, potentially resulting in DNA hypomethylation. *NSD1* inhibition has been shown to enhance sensitivity to cisplatin and carboplatin in head and neck cancer cell lines [47,48], but the impact of loss-of-function mutations in *NSD1* on cetuximab response is unknown.

The Ctx$^R$ 4 cells bear unique mutations in *PIK3R2* and *RB1* (**Table 2**). *PIK3R2* encodes p85β, a regulatory subunit of PI3K that is rarely mutated in HNSCC (~1%) [6,49–51]. The functional significance of the non-frameshift substitution (c.700_702CGT) identified in the Ctx$^R$ 4 cells is unknown; however, in a study focusing on mutations in *PIK3R2* and other PI3K pathway-associated genes in endometrial cancer, it was suggested that *PIK3R2* mutations may phenocopy loss of the tumor suppressor phosphatase and tensin homolog (PTEN) [49], an alteration that leads to aberrant hyperactivation of PI3K/Akt signaling, which has previously been implicated in cetuximab resistance [3,14,22,26]. The presence of a stopgain mutation (p. W563*) in *RB1* (which encodes retinoblastoma protein) suggests that this tumor suppressor may be functionally inactivated in the Ctx$^R$ 4 cells. As the CDK4/6 inhibitor palbociclib was recently shown to act synergistically with either lapatinib or afatinib (tyrosine kinase inhibitors that target EGFR and HER2) to inhibit proliferation in HNSCC cell lines [52], it may be worthwhile to test whether co-treatment of the Ctx$^R$ 4 cells with palbociclib has an impact on their response to cetuximab.

In light of a recent publication focusing on the timing of alterations developed during the acquisition of cetuximab resistance in an HNSCC cell line [53], the mutations in genes associated with epigenetic regulation, such as *KAT6A* and *NSD1*, identified in the targeted sequencing analysis may be of particular interest. In the aforementioned study, published by Stein-O'Brien and colleagues, a rigorous time course analysis consisting of weekly collection of samples for RNA-sequencing and DNA methylation analyses revealed that, while transcriptional changes arose quickly following initiation of cetuximab treatment, stable alterations in DNA methylation were observed only after resistance was established, highlighting a difference between adaptive responses to treatment and acquired resistance. In addition, we recently

reported that the chromatin reader protein BRD4 promotes cetuximab resistance in HNSCC cell line models, including the PE/CA-PJ49 CtxR cells used in this study [30]. Studying epigenetic alterations in cetuximab-resistant cells, whether or not these alterations are related to the mutations detected in our targeted sequencing analysis, may identify additional mechanisms of cetuximab resistance and potential candidates to target to prevent and/or overcome cetuximab resistance.

IL-6 upregulation has been repeatedly demonstrated in the context of treatment with, and resistance to, cetuximab and other EGFR-targeted therapies [13,17,43,44]. Upon observation that IL-6 secretion was increased in PE/CA-PJ49 CtxR cells compared to parental cells, we expected to find that IL-6 mediated cetuximab resistance and that inhibiting the IL-6 pathway would restore cetuximab sensitivity in CtxR cells. Instead, we found that the increase in IL-6 secretion by PE/CA-PJ49 CtxR cells belied a functional impairment in the IL-6 signaling pathway. These findings demonstrate that, even when IL-6 levels are increased in the context of cetuximab resistance, this does not necessarily indicate that IL-6 mediates cetuximab resistance. This study highlights the importance of differentiating between alterations that are simply correlated with cetuximab resistance and those that play a functional role in maintaining cetuximab resistance in order to identify promising candidates to target to overcome cetuximab resistance.

## Materials and methods

### Cell culture

Cell lines were maintained in Dulbecco's Modified Eagle's Medium (DMEM; Corning 10-013-CM) containing 10% fetal bovine serum (FBS; Gemini Bio-Products #900–108) and penicillin-streptomycin (Gibco 15140–122). PE/CA-PJ49 cells were purchased from Sigma-Aldrich. Cell lines were authenticated by short tandem repeat (STR) analysis performed by the University of California, Berkeley DNA Sequencing Facility at least once every 6 months.

### Generation of cetuximab-resistant cell lines

We previously reported generation of the PE/CA-PJ49 cell line models of acquired cetuximab resistance [30]. STR analysis was performed on the cetuximab-resistant cell lines to confirm that the profiles matched those of the parental cell line.

### Dose-response assays

Cells were plated in 96-well plates and incubated overnight. The next day, the media was replaced with media containing the indicated concentrations of drug or an equivalent volume of vehicle. After 96 hours of treatment, cells were rinsed with ice-cold phosphate-buffered saline (PBS) and stained with crystal violet solution (0.5% crystal violet [Sigma C0775] in 25% methanol). To quantify results, crystal violet-stained cells were solubilized in a 1:1 mixture of 200 mM sodium citrate and 100% ethanol and absorbance at 590 nm was read using a Biotek Epoch Microplate Spectrophotometer.

### Gene expression analysis

RNA was isolated from HNSCC cells using the RNeasy Mini Kit (Qiagen, 74106), according to the manufacturer's protocol (excluding the optional DNase digestion step) and eluted in nuclease-free water (Fisher BioReagents, BP2484-50). RNA concentration and purity (OD 260/280) were determined using the Biotek Epoch Microplate Spectrophotometer. One microgram of RNA per sample was converted to complementary DNA (cDNA) in an Eppendorf PCR

machine using either iScript™ Reverse Transcription Supermix for RT-qPCR (Bio-Rad, #1708840) or iScript™ cDNA Synthesis Kit (Bio-Rad, #1708890), according to the manufacturer's instructions. Quantitative reverse transcription PCR (qPCR) was performed in the CFX96 Touch™ Real-Time PCR Detection System (Bio-Rad) using the iTaq™ Universal SYBR® Green Supermix (Bio-Rad, #1725124) according to the manufacturer's instructions. Assay of each sample was performed in technical duplicate. The sequences of the primers used are listed in S1 Table. The delta-delta Ct method was used to determine relative mRNA expression (normalized to the reference gene TATA-box-binding protein [*TBP*]). GraphPad Prism was used to conduct Student's two-tailed t-test to determine whether changes between experimental conditions were statistically significant. Controls for each experiment and number of biological replicates are indicated in the respective figure legends.

## Immunoblot analysis

Cells for immunoblot analysis were rinsed twice with ice-cold PBS and lysed on ice in radioimmunoprecipitation assay (RIPA) buffer (150 mM sodium chloride, 5 mM EDTA pH 8.0, 50 mM Tris pH 8.0, 1% NP-40 Surfact-Amps Detergent Solution, 0.5% sodium deoxycholate, 0.1% sodium dodecyl sulfate [SDS]) supplemented with cOmplete Protease Inhibitor Cocktail (Roche 11836145001) and PhosSTOP Phosphatase Inhibitor Cocktail Tablets (Roche 04 906 837 001). Cells undergoing lysis were briefly vortexed, then centrifuged for 10 minutes at 13,200 RCF at 4°C. Supernatants were transferred to fresh 1.7-mL microcentrifuge tubes and protein concentrations were determined using Protein Assay Dye Reagent (Bio-Rad #5000006). Lysates were mixed with the appropriate volume of 4X sample buffer (230 mM Tris-HCl pH 6.8, 7% SDS, 32% glycerol, 0.1% w/v bromophenol blue, 9% β-mercaptoethanol), boiled for 5 minutes, electrophoresed on 10% polyacrylamide Tris-glycine gels, and transferred to Immun-Blot PVDF membranes (Bio-Rad #1620177) on a Trans-Blot SD Semi-Dry Transfer Cell (Bio-Rad). Membranes were then blocked in 5% nonfat dry milk (Apex 20–241) in Tris-buffered saline with Tween 20 (TBST) and probed overnight at 4°C with primary antibodies diluted in 2.5% bovine serum albumin (BSA; Sigma A3912) in TBST. Primary antibodies used in this study were purchased from Cell Signaling Technology (P-STAT3^Y705 [#9145, rabbit monoclonal] and STAT3 [#4904, rabbit monoclonal]), Santa Cruz Biotechnology (gp130 [sc-376280, Lot #B1717, mouse monoclonal]), and Abcam (β-tubulin [ab6046, rabbit polyclonal]).

The next day, membranes were washed 5–6 times in TBST, blocked for 15–30 min in 5% nonfat dry milk in TBST, and incubated in the appropriate horseradish peroxidase (HRP)-conjugated secondary antibody for 45–90 min at room temperature. Secondary antibodies used in this study were purchased from Bio-Rad (Goat Anti-Rabbit IgG (H+L)-HRP Conjugate [#1706515] and Goat Anti-Mouse IgG (H+L)-HRP Conjugate [#1706516]). After incubation in secondary antibody, membranes were washed 5–6 times in TBST and incubated in the chemiluminescent HRP substrate Western Blotting Luminol Reagent (Santa Cruz Biotechnology, sc-2048) according to the manufacturer's instructions. Films (GeneMate F-9024-8X10) were scanned at 300 dpi and images were converted to greyscale prior to densitometric analysis using ImageJ (National Institutes of Health), but were not otherwise altered. Density values for the proteins of interest were divided by those of the total protein (for phosphorylated proteins) or the loading control (β-tubulin) (for all other proteins) from the corresponding lane of the same membrane. Data were normalized by dividing the values for each sample by the average of those for the control samples (controls for each experiment and number of biological replicates are indicated in the respective figure legends). GraphPad Prism was used to conduct Student's two-tailed t-test to determine whether changes between experimental conditions were statistically significant.

## Enzyme-linked immunosorbent assay (ELISA)

Cells were plated at 50,000 cells per well in 24-well culture plates in DMEM supplemented with 10% FBS and penicillin/streptomycin and allowed to attach overnight. The next day, cells were gently rinsed with sterile PBS and media was replaced with 500 μL DMEM (without FBS or penicillin-streptomycin). After 72 hours of incubation, conditioned media were removed and centrifuged for 10 min at 10,000 RCF at 4˚C. Supernatants were transferred to fresh 1.7-mL microcentrifuge tubes and frozen at -80˚C for subsequent analysis. The concentration of IL-6 in the cell culture supernatants was determined using the Human IL-6 DuoSet ELISA kit (R&D Systems DY206) according to the manufacturer's instructions. Plates were read at 450 nm on the Biotek Epoch Microplate Spectrophotometer, with wavelength correction at 540 nm. Assays were performed in technical duplicate. The number of biological replicates is indicated in the respective figure legends. GraphPad Prism was used to conduct Student's two-tailed t-test to determine whether changes in concentration of secreted IL-6 were statistically significant between parental PE/CA-PJ49 cells and PE/CA-PJ49 cells that had acquired resistance to cetuximab.

## Inhibitors and cytokines

Erlotinib (S1023), afatinib (S7810), and lapatinib (S2111) were purchased from Selleckchem. Cisplatin was purchased from the University of Pittsburgh Cancer Institute Pharmacy. CBL0137 was provided by Dr. George Stark and Dr. Sarmishtha De (Cleveland Clinic). JQ1 was provided by Dr. James Bradner (Dana-Farber Cancer Institute, Boston, MA). Recombinant cytokines used in this study were purchased from PeproTech (Recombinant Human IL-6 [200–06], Recombinant Human LIF [300–05], and Recombinant Human Oncostatin M (209 a.a.) [300-10T]). Lyophilized cytokines were reconstituted in sterile nuclease-free water (Fisher BioReagents, BP2484-50) before use.

## siRNA transfection

Cells were plated in DMEM containing 10% FBS and penicillin-streptomycin and allowed to attach overnight. The next day, immediately prior to transfection, media was replaced with DMEM supplemented with 10% FBS (without antibiotics). A final concentration of 10 nM siRNA (control [nontargeting] siRNA or one of at least two distinct siRNA sequences per target) was transfected into cells using Lipofectamine™ RNAiMAX Transfection Reagent (ThermoFisher Scientific, #13778500) according to the manufacturer's instructions, using 5 μL RNAiMAX in a total volume of 1.5 mL for 6-well plates and 2.5 μL RNAiMAX in a total volume of 750 μL for 12-well plates. The siRNA-containing media was replaced with DMEM supplemented with 10% FBS and penicillin-streptomycin 4 hours post-transfection. Knockdown was validated using qPCR analysis. All siRNAs were purchased from Origene (IL6 Human siRNA Oligo Duplex [SR302379], IL6R Human siRNA Oligo Duplex [SR302380], and IL6ST Human siRNA Oligo Duplex [SR302381]).

## Clonogenic survival assays

Cells were plated at 250 cells per well in 12-well cell culture plates. The next day, cells were treated as indicated for the particular experiment. Media containing vehicle and/or drug was replaced every four days. After 12 days of treatment, cells were stained with crystal violet solution (0.5% crystal violet [Sigma C0775] in 25% methanol).

## UCSF500 Cancer Gene Panel

Genomic DNA was extracted from the PE/CA-PJ49 parental and Ctx$^R$ cells using the Qiagen DNeasy Blood & Tissue kit (Catalog number 69504) according to the manufacturer's instructions, then submitted to the UCSF Clinical Cancer Genomics Laboratory for testing using the UCSF500 Cancer Gene Panel. The UCSF 500 Cancer Gene Panel uses capture-based next-generation sequencing to target and analyze the coding regions (exons) of 479 cancer genes, as well as select introns of 47 genes. Target enrichment was performed by hybrid capture using custom oligonucleotides (Roche Nimblegen). Sequencing of captured libraries was performed on an Illumina HiSeq 2500 in rapid run mode (2 X 101 bp read length). Sequence reads were de-duplicated to allow for accurate allele frequency determination and copy number calling. The analysis used open source or licensed software for alignment to the human reference sequence UCSC build hg19 (NCBI build 37) and variant calling. Common germline polymorphisms were eliminated from analysis using the complete list of germline variants from dbSNP. Rare variants were reviewed by using a filtering threshold of 0.1% in large population databases (gnomAD; https://gnomad.broadinstitute.org/). Additional filtering to eliminate technology specific sequencing artifacts was performed before analyzing the data.

## Supporting information

**S1 Fig. BET inhibition reduces *IL6* expression in PE/CA-PJ49 parental and Ctx$^R$ cells.** PE/CA-PJ49 parental, Ctx$^R$ 3, and Ctx$^R$ 4 cells were treated with vehicle (DMSO) or 300 nM JQ1. After 96 hours of treatment, RNA was extracted and qPCR was conducted using the *IL6* primers listed in **S1 Table** (normalized to *TBP)*. $n$ = 3. $^*$p$<$0.05; $^{***}$p$<$0.001.
(TIF)

**S2 Fig. Cetuximab-resistant PE/CA-PJ49 cells are cross-resistant to EGFR-targeted TKIs. A, B, C)** Erlotinib **(A)**, afatinib **(B)**, and lapatinib **(C)** dose response assays in PE/CA-PJ49 parental cells and Ctx$^R$ clones treated for 96 h, then stained with crystal violet. $n$ = 6. **D)** PE/CA-PJ49 parental cells and Ctx$^R$ clones were plated at low density and treated with vehicle (DMSO), 100 nM erlotinib, 1 nM afatinib, or 1 µM lapatinib, then stained with crystal violet after 12 days of treatment. Media containing vehicle or drug was changed every four days. $n$ = 4.
(TIF)

**S3 Fig. Cetuximab-resistant PE/CA-PJ49 cells are not resistant to cisplatin and CBL0127. A, B)** Cisplatin **(A)** and CBL0137 **(B)** dose response assays in PE/CA-PJ49 parental and Ctx$^R$ cells treated for 96 h, then stained with crystal violet. $n$ = 6.
(TIF)

**S4 Fig. siRNA-mediated knockdown of *IL6R* does not impact cetuximab response in PE/CA-PJ49 parental and Ctx$^R$ cells. A)** PE/CA-PJ49 parental cells were transfected with 10 nM nontargeting (nt) siRNA or one of two siRNAs targeting *IL6R* (siIL6R A and C). RNA was extracted 96 hours post-transfection and qPCR was conducted using the *IL6R* primers listed in **S1 Table** (normalized to *TBP)*. $n$ = 3. $^{**}$p$<$0.01. **B)** PE/CA-PJ49 parental and Ctx$^R$ cells were plated at a low density and transfected with 10 nM siRNA the next day. On the following day, and every four days thereafter, the cells were treated with vehicle (PBS) or 100 nM Ctx. The cells were stained with crystal violet 13 days post-transfection.
(TIF)

**S5 Fig. siRNA-mediated knockdown of *IL6ST* does not impact cetuximab response in PE/CA-PJ49 parental and Ctx$^R$ cells. A)** PE/CA-PJ49 parental cells were transfected with 10 nM

nontargeting (nt) siRNA or one of three siRNAs targeting *IL6ST* (siIL6ST A, B, and C). RNA was extracted 96 hours post-transfection and qPCR was conducted using the *IL6ST* primers listed in **S1 Table** (normalized to *TBP).* n = 3. ****p<0.0001. **B)** PE/CA-PJ49 parental and Ctx<sup>R</sup> cells were plated at a low density and transfected with 10 nM siRNA the next day. On the following day, and every four days thereafter, the cells were treated with vehicle (PBS) or 100 nM Ctx. The cells were stained with crystal violet 13 days post-transfection.
(TIF)

**S6 Fig. Pharmacological inhibition of the IL-6 pathway does not impact cetuximab response in PE/CA-PJ49 parental and Ctx<sup>R</sup> cells. A)** Serum-starved PE/CA-PJ49 parental cells were pre-treated for 2 hours with vehicle (PBS) or 100 nm– 5 μM TCZ, then treated with 50 ng/mL rhIL6 for 15 minutes. Cells were lysed in RIPA buffer and immunoblot was performed. β-tubulin image shown is from the STAT3 blot. **B)** PE/CA-PJ49 parental and Ctx<sup>R</sup> cells were plated at a low density, then treated with vehicle (PBS), 100 nM Ctx, 1 μM TCZ, or the combination of Ctx and TCZ every 4 days. After a total of 12 days of treatment, the cells were stained with crystal violet. **C)** Crystal violet-stained cells from **(B)** were solubilized and absorbance at 590 nm was measured. Student's two-tailed t-test was used to determine whether differences in absorbance at 590 nm were statistically significant (compared to vehicle-treated cells). *n* = 3. *p<0.05; **p<0.01.
(TIF)

**S7 Fig. Phosphorylation of STAT3 is induced in PE/CA-PJ49 Ctx<sup>R</sup> cells treated with rhOSM, but not rhIL6.** PE/CA-PJ49 parental and Ctx<sup>R</sup> cells were serum starved for 4 hours, then treated for 15 minutes with 50 ng/mL rhIL6 or rhOSM. Cells were lysed in RIPA buffer and immunoblot was performed.
(TIF)

**S8 Fig. *IL6* mRNA expression is increased in Ctx-treated PE/CA-PJ49 parental cells.** PE/CA-PJ49 parental cells were treated with vehicle (PBS) or 100 nM Ctx. After 96 hours of treatment, RNA was extracted and qPCR was conducted using the *IL6* primers listed in **S1 Table** (normalized to *TBP).* Student's two-tailed t-test was used to determine whether differences in *IL6* expression were statistically significant. *n* = 3. **p<0.01.
(TIF)

**S1 Table. qPCR primers.**
(DOCX)

**S2 Table. UCSF500 results–PE/CA-PJ49 parental cells.**
(XLSX)

**S3 Table. UCSF500 results–PE/CA-PJ49 Ctx<sup>R</sup> 1 cells.**
(XLSX)

**S4 Table. UCSF500 results–PE/CA-PJ49 Ctx<sup>R</sup> 3 cells.**
(XLSX)

**S5 Table. UCSF500 results–PE/CA-PJ49 Ctx<sup>R</sup> 4 cells.**
(XLSX)

**S1 Raw Images. Original blot images.**
(PDF)

## Acknowledgments

We gratefully acknowledge Dr. George Stark and Dr. Sarmishtha De (Cleveland Clinic) for providing CBL0137 and the UCSF Clinical Cancer Genomics Laboratory, especially Dr. Jessica Van Ziffle, for performing the UCSF500 sequencing and providing guidance with data analysis.

## Author Contributions

**Conceptualization:** Rachel A. O'Keefe, Jennifer R. Grandis.

**Funding acquisition:** Daniel E. Johnson, Jennifer R. Grandis.

**Investigation:** Rachel A. O'Keefe, David S. Lee.

**Methodology:** Neil E. Bhola.

**Supervision:** Jennifer R. Grandis.

**Visualization:** Rachel A. O'Keefe.

**Writing – original draft:** Rachel A. O'Keefe.

**Writing – review & editing:** Neil E. Bhola, David S. Lee, Daniel E. Johnson, Jennifer R. Grandis.

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
