## [Decision Letter · Decision Letter 0]

15 Oct 2019

PONE-D-19-25508

Interleukin 6 is increased in preclinical HNSCC models of acquired cetuximab resistance, but is not required for maintenance of resistance

PLOS ONE

Dear Ms. O'Keefe,

Thank you for submitting your manuscript to PLOS ONE. After careful consideration, we feel that it has merit but does not fully meet PLOS ONE’s publication criteria as it currently stands. Therefore, we invite you to submit a revised version of the manuscript that addresses the points raised during the review process.

We would appreciate receiving your revised manuscript by Nov 29 2019 11:59PM. To enhance the reproducibility of your results, we recommend that if applicable you deposit your laboratory protocols in protocols.io, where a protocol can be assigned its own identifier (DOI) such that it can be cited independently in the future. For instructions see: http://journals.plos.org/plosone/s/submission-guidelines#loc-laboratory-protocols

We look forward to receiving your revised manuscript.

Kind regards,

Karl X Chai, Ph.D.

Academic Editor

PLOS ONE

Journal Requirements:

DEJ and JRG are co-inventors of cyclic STAT3 decoy and have financial interests in STAT3 Therapeutics. STAT3 Therapeutics holds an interest in cyclic STAT3 decoy, which was not used in the studies in this manuscript. The remaining authors declare no conflicts.

Reviewers' comments:

Reviewer's Responses to Questions

**Comments to the Author**

1. Is the manuscript technically sound, and do the data support the conclusions?

Reviewer #1: Yes

2. Has the statistical analysis been performed appropriately and rigorously? 

Reviewer #1: Yes

3. Have the authors made all data underlying the findings in their manuscript fully available?

Reviewer #1: Yes

4. Is the manuscript presented in an intelligible fashion and written in standard English?

Reviewer #1: Yes

5. Review Comments to the Author

Reviewer #1: This paper by O’Keefe et al.: “Interleukin 6 is increased in preclinical HNSCC models of acquired cetuximab resistance, but is not required for maintenance of resistance” is a detailed examination of the potential mechanisms for IL-6 in acquired cetuximab resistence, using a cetuximab sensitive oral cancer cell line and sister clone that has been made resistant in vitro (CTxR clone PE/CA-PJ49). The resistant clone produce increased amount of IL-6. However, addition of exogenous IL-6 to the sensitive clone did not make them cetuximab resistance, and IL-6 pathway inhibition did not restore cetuximab sensitivity in the resistant cells, presumably because the IL6R expression was reduced which may have explained why IL-6 treatment did not induce STAT3 phosphorylation in the resistant cells. The authors concluded that even if IL-6 is increased in the cetuximab resistance cell line, it was not responsible for resistance, so targeting the IL-6 pathway as STAT3 inhibition, may not restore cetuximab sensitivity in resistant HNSCC.

In search for genetic alterations that may explain the acquired cetuximab resistance, the authors used the next-generation DNA sequencing using the UCSF500 Cancer gene panel version 2, that sequence approximately 500 cancer related genes. The authors presents filtered data in two tables, one showing mutation common for all the PECA-PJ49 subclones, and some, but presumably not all genes that are selective mutated in each of the clones. The unfiltered data is provided as supplementary files (S2-S5).

The paper is generally well-written, -documented and -illustrated. It contains novel information on how upregulated IL6 in resistant PJ49-subclones did not mediate cetuximab resistance. This is a refreshingly honest, basically a negative scientific report from a group that has a particular interest in STAT3 inhibition as treatment for HNSCC. Although this has been examined in one cell line, only, it may be a more generalized phenomenon in acquired resistance, as similar findings had been found in acquired cisplatin-resistance HNSCC cell lines from another group. However there are some point which needs to be considered.

1. This group have made at least 3 HNSCC cell lines cetuximab resistant, PECAPJ49, FaDu, and CAL33 cells, as previously reported. Was the IL-6 increase a sole PECAPJ49 cell line phenomenon, or did all resistant cell lines express increased IL-6? Please include a comment on the other cetuximab resistant cell lines.

2. The few mutated genes unique for the subclones, were these the only one that were unique?

3. One of the mutation in CtxR1 is KATA6A, a Histone Acetyltransferase (H3K23) that through TRIM24 activate PIK3CA transcription, thereby enhancing PI3K/AKT signaling and tumorigenesis. Does this explain the IL-6 increase?

4. When targeting BRD44, which restored cetuximab sensitivity in the cetuxamab resistent HNSCC cell lines (previous paper), did the IL-6 level decrease?

5. Some more information on the other subclone-specific mutations (NSD1, CHD5, RB1 etc) would be appreciated.

6. Is it possible to understand the cetuximab resistance from these gene alterations? There is no discussion or explanation to whether these or other gene alteration may explain the resistance.

7. The supplementary data of the unfiltered UCSF500 Cancer gene panel data, there are a generally lack of explanation and legends. In particular S3 appear be the list of genes included in the examinations, rather than one of the results as included in S2 and S4-5, with no explanation.

8. It is also difficult to understand the rationale behind the filtrations as it is not stated. For most of the readers that do not work with these data, there may be a need for å instruction for what the different columns in the raw data stand for, and which of the files belong to which subclone. The short title of the columns do not say much and the home page for the UCSF cancer panel did not give any link to a better explanation.

9. The authors discuss how mutation in HRAS have been implicated in cetuximab resistance (line 71-71 in the introduction and refereeing to ref: 22, 23). The HRAS is not mutated neither in the original PECAPJ49 cell clone nor in the resistant subclones. However, this cell line do carries several RAS-associated mutations (RASA1, RASA2, etc). Do these mutations influence how easy the cell line become resistant?

10. Minor point. The Ctx abbreviation for cetuximab has actually not been stated in the manuscript prior to its use (line 215).

6. PLOS authors have the option to publish the peer review history of their article (what does this mean?). If published, this will include your full peer review and any attached files.

Reviewer #1: Yes: Trond S. Halstensen

---

## [Author Response · Author response to Decision Letter 0]

30 Nov 2019

1. This group have made at least 3 HNSCC cell lines cetuximab resistant, PECAPJ49, FaDu, and CAL33 cells, as previously reported. Was the IL-6 increase a sole PECAPJ49 cell line phenomenon, or did all resistant cell lines express increased IL-6? Please include a comment on the other cetuximab resistant cell lines.

We also observed an increase in IL6 mRNA expression in the other two cetuximab-resistant HNSCC cell lines, FaDu and Cal33 cells, compared to the parental FaDu and Cal33 cells, respectively. We elected to focus on the PE/CA-PJ49 models because FaDu cells do not express the receptor subunit gp130, which is required for IL-6 signal transduction, and because, though IL6 mRNA expression was slightly increased in the Cal33 CtxR models compared to parental Cal33 cells, IL-6 levels in the Cal33 parental and CtxR cell culture supernatants were very low (~25-50 pg/mL). We have added this information to the manuscript (lines 123-130).

2. The few mutated genes unique for the subclones, were these the only one that were unique?

Yes, the mutated genes listed as unique to each of the clones were the only single nucleotide variants and indels identified in the UCSF500 analysis (once the data were filtered to exclude common polymorphisms). The manuscript has been updated to include this information (lines 357-358).

3. One of the mutation in CtxR1 is KAT6A, a Histone Acetyltransferase (H3K23) that through TRIM24 activate PIK3CA transcription, thereby enhancing PI3K/AKT signaling and tumorigenesis. Does this explain the IL-6 increase?

We thank the reviewer for drawing our attention to the prior study linking KAT6A to PI3K/Akt pathway activation. Because KAT6A is altered in only the CtxR 1 cells, we suspect that another mechanism(s) may be driving increased IL6 expression in the CtxR cells. However, because activation of the PI3K/Akt signaling pathway has previously been implicated in cetuximab resistance, we have added a discussion of a possible role for KATA6A-induced upregulation of PIK3CA in cetuximab resistance in these models (lines 477-482 in marked-up copy; lines 470-477 in clean copy).

4. When targeting BRD4, which restored cetuximab sensitivity in the cetuximab resistant HNSCC cell lines (previous paper), did the IL-6 level decrease?

We appreciate this question and have addressed it by treating the PE/CA-PJ49 cells with JQ1, which reduced IL6 mRNA expression. We have added these data as a supplemental figure (S1 Fig) and updated the text to describe these results (lines 123-125).

5. Some more information on the other subclone-specific mutations (NSD1, CHD5, RB1 etc) would be appreciated. and 6. Is it possible to understand the cetuximab resistance from these gene alterations? There is no discussion or explanation to whether these or other gene alteration may explain the resistance.

We apologize for this oversight and have expanded the discussion of the mutations unique to each subclone to include more information about the functions of the proteins encoded by these genes and hypotheses about how mutations in these genes may contribute to cetuximab resistance (lines 470-504 in marked-up copy; lines 465-499 in clean copy).

7. The supplementary data of the unfiltered UCSF500 Cancer gene panel data, there are a generally lack of explanation and legends. In particular S3 appear be the list of genes included in the examinations, rather than one of the results as included in S2 and S4-5, with no explanation. and 8. It is also difficult to understand the rationale behind the filtrations as it is not stated. For most of the readers that do not work with these data, there may be a need for å instruction for what the different columns in the raw data stand for, and which of the files belong to which subclone. The short title of the columns do not say much and the home page for the UCSF cancer panel did not give any link to a better explanation.

We thank the reviewer for these suggestions to improve clarity and facilitate future analysis by other researchers. We have expanded the column titles and simplified the spreadsheets. In addition, we have modified the Materials and Methods section to explain the rationale for data filtration (lines 662-673 in marked-up copy; lines 654-665 in clean copy).

9. The authors discuss how mutation in HRAS have been implicated in cetuximab resistance (line 71-71 in the introduction and refereeing to ref: 22, 23). The HRAS is not mutated neither in the original PECAPJ49 cell clone nor in the resistant subclones. However, this cell line do carries several RAS-associated mutations (RASA1, RASA2, etc). Do these mutations influence how easy the cell line become resistant?

This is an intriguing hypothesis, and with RASA1 and RASA2 alterations occurring in 4% and 12% of HNSCC tumors, respectively, in the TCGA provisional dataset on cBioPortal, determining whether alterations in RASA1 and RASA2 promote intrinsic and/or acquired cetuximab resistance may be a topic of future study. At this time, however, we do not have evidence implicating these RAS-associated genes in the acquisition of cetuximab resistance in these models.

10. Minor point. The Ctx abbreviation for cetuximab has actually not been stated in the manuscript prior to its use (line 215).

We thank the reviewer for pointing this out and have included the abbreviation for cetuximab the first time it appears in the main text (line 57).

---

## [Decision Letter · Decision Letter 1]

17 Dec 2019

Interleukin 6 is increased in preclinical HNSCC models of acquired cetuximab resistance, but is not required for maintenance of resistance

PONE-D-19-25508R1

Dear Dr. O'Keefe,

We are pleased to inform you that your manuscript has been judged scientifically suitable for publication and will be formally accepted for publication once it complies with all outstanding technical requirements.

With kind regards,

Karl X Chai, Ph.D.

Academic Editor

PLOS ONE

Additional Editor Comments (optional):

Reviewers' comments:

Reviewer's Responses to Questions

**Comments to the Author**

1. If the authors have adequately addressed your comments raised in a previous round of review and you feel that this manuscript is now acceptable for publication, you may indicate that here to bypass the “Comments to the Author” section, enter your conflict of interest statement in the “Confidential to Editor” section, and submit your "Accept" recommendation.

Reviewer #1: All comments have been addressed

2. Is the manuscript technically sound, and do the data support the conclusions?

Reviewer #1: Yes

3. Has the statistical analysis been performed appropriately and rigorously? 

Reviewer #1: Yes

4. Have the authors made all data underlying the findings in their manuscript fully available?

Reviewer #1: Yes

5. Is the manuscript presented in an intelligible fashion and written in standard English?

Reviewer #1: Yes

6. Review Comments to the Author

Reviewer #1: Minor point, line 477, : " NSD1 encodes nuclear receptor binding SET domain protein 1 (NSD1), a histone methyltransferase...."

Remove one of the abrivations, no need for two "NSD1" (or

7. PLOS authors have the option to publish the peer review history of their article (what does this mean?). If published, this will include your full peer review and any attached files.

Reviewer #1: Yes: Trond S. Halstensen

---

## [Editor Report · Acceptance letter]

30 Dec 2019

PONE-D-19-25508R1 

Interleukin 6 is increased in preclinical HNSCC models of acquired cetuximab resistance, but is not required for maintenance of resistance 

Dear Dr. O'Keefe:

I am pleased to inform you that your manuscript has been deemed suitable for publication in PLOS ONE. Congratulations! Your manuscript is now with our production department. 

With kind regards,

on behalf of

Dr. Karl X Chai 

Academic Editor

PLOS ONE